# FemoraLyze: A Modular Framework
# for Proximal Femur Analysis

**Marten J. Finck**[*1] (iD)                    MAFI@INFORMATIK.UNI-KIEL.DE
[1] *Visual Computing and Artificial Intelligence, Kiel University, Kiel*
**Niklas C. Koser**[*2] (iD)                    NIKLAS.KOSER@RAD.UNI-KIEL.DE
**Jan-Bernd Hövener**[2] (iD)                    JAN.HOEVENER@RAD.UNI-KIEL.DE
**Claus-Christian Glüer**[2] (iD)                    GLUEER@RAD.UNI-KIEL.DE
[2] *i2Lab@Section Biomedical Imaging, Kiel University, University Hospital Schleswig-Holstein, Kiel*
**Sören Pirk**[1] (iD)                    SP@INFORMATIK.UNI-KIEL.DE

**Editors:** Accepted for publication at MIDL 2025

## Abstract

The proximal femur is exposed to an increased risk of fracture, particularly in the context of osteoporosis. As the prevalence increases with age, it is expected that the incidence of osteoporotic fractures will likely continue to rise in terms of demographic trends. Early, guideline-based therapy offers strong prospects of success, but requires precise and reliable diagnostic and prognostic procedures. Automated bone metrics are suitable for this purpose and could also be used for other applications such as preoperative planning of total hip arthroplasty in patients affected by arthritis. In this paper we present FemoraLyze, which is a modular deep-learning-based Python framework that combines the automated and differentiated calculation of segmentation masks, bone structure and geometry parameters of the proximal femur based on a computed tomography (CT) image. The code is available at https://github.com/martenfi/FemoraLyze.

**Keywords:** Femur Assessment, Deep Learning, Bone Microstructure, Bone Geometry

## 1. Introduction

The proximal femur is subject to continuous biomechanical stress over the course of life, which leads to structural changes in the bone tissue (Ito et al., 2011). Thus, with increasing age, the incidence of osteoporosis and arthritis increases significantly: while osteoporosis is primarily associated with increased fragility and an increased risk of fracture, arthritis primarily leads to chronic joint pain (Deng et al., 2022; Anderson and Loeser, 2010). Given demographic trends and an aging population, a growing number of individuals are expected to experience reduced quality of life due to the aforementioned diseases, resulting in a substantial burden on the healthcare systems (Kanis et al., 2021). To address this, clinically relevant metrics of the proximal femur are essential for effective prognosis (Herbst et al., 2021; Deng et al., 2022; Huppke et al., 2024), diagnosis (Poole et al., 2012), and therapy (Wei et al., 2020; Macdonald et al., 2011). For example, promising preventive measures can be initiated (Deng et al., 2022; Herbst et al., 2021; Neeteson et al., 2023), pathologies can be quantified in terms of their severity and progression (Macdonald et al., 2011), and

---

[*] Contributed equally

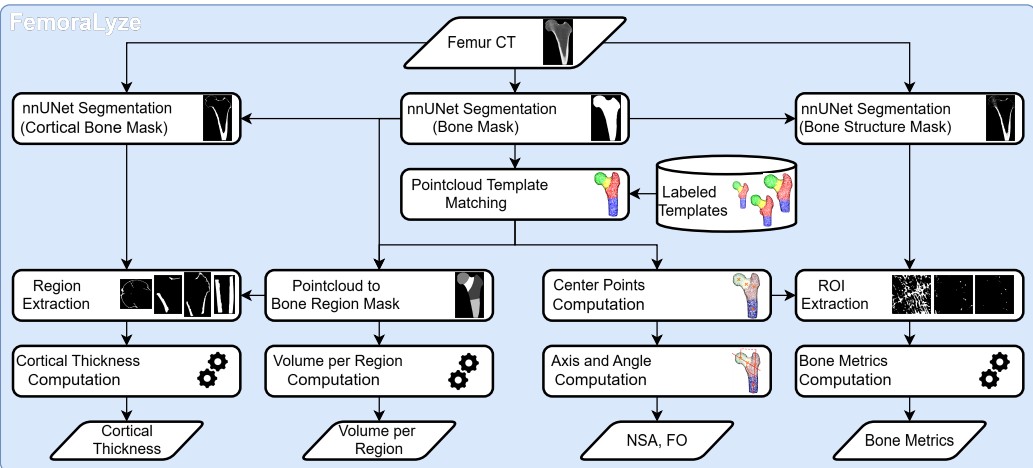

Figure 1: Structure and data flows of the modular framework FemoraLyze.

precise preoperative planning (Li et al., 2022) can be enabled. In this paper, we present FemoraLyze, a modular framework for the fully-automated analysis of the proximal femur. It consists of several processing modules which can be flexibly adapted or extended. Based on CT data of the femur, FemoraLyze uses deep learning (DL) models to extract various segmentations. These are used for the generation of point clouds, the identification of characteristic regions of interest (ROIs), the calculation of geometric bone properties and the extraction of bone structure metrics.

## 2. Methods and Results

**Dataset.** This study primarily uses 50 HR-pQCT images (XtremeCT, SCANCO Medical AG) of the left proximal femur acquired *ex situ* with an isotropic spacing of 82 μm. The data set includes male and female individuals of different ages, heights and weights. The study was approved by the ethics committee of the Hamburg chamber of physicians (WF-057/21). **Framework.** Figure 1 illustrates the structure and data flow of FemoraLyze. Three nnU-NetV 2 models were trained as a basis for the following analyses (Isensee et al., 2021): (1) A model for the extraction of the cortical bone, trained on 11 (masked) CTs with cortical masks created manually using nnInteractive (Isensee et al., 2025); (2) A model for segmentation of all bone structures, based on 11 (masked) CTs with masks generated by thresholding; (3) Another model trained on 12 downsampled CTs with binary bone masks generated by a nnU-Net pre-trained on 142 femora of the TotalSegmentator dataset (Wasserthal et al., 2023). A suitable template is selected based on the bone volume of the bone mask. After generating a surface point cloud, registration is performed in three steps: center-point alignment, RANSAC and ICP with scaling. The labels of the template are then transferred to the point cloud. Center points for four anatomical regions are calculated from the labeled point cloud, which are used to determine the neck-shaft angle (NSA) and the femoral offset (FO) (Yang et al., 2023). They also enable the extraction of ROIs from the bone structure mask for the calculation of trabecular parameters (BV/TV, Tb.Th, Tb.Sp, Tb.N) using the Hildebrand algorithm (Hildebrand and Rüegsegger, 1997). A bone region mask is generated by filling the point cloud regions and applying a closest point algorithm. This is

used for volumetric analysis and regionally differentiated extraction from the cortical mask to determine the cortical thickness (Ct.Th) analogous to Tb.Th.

**Experiments.** Since all quantitative evaluations are based on the segmentation masks generated by the nnU-Nets and only these modules enable ground truth generation without medical expertise, the validation is limited to the bone mask, the bone structure mask and the cortical bone mask. Three independent reference masks that are not used in the training process are used for objective evaluation (generated with nnInteractive (Isensee et al., 2025) and 3D Slicer (Fedorov et al., 2012)).

**Results.** The nnU-Net for cortical bone segmentation achieved a mean Dice score of 0.94 and an mean IoU of 0.88 on three test bones. The model for segmenting the bone structure achieved a Dice score of 0.93 and an IoU of 0.86, while the model for predicting the bone mask achieved a Dice score of 0.95 and an IoU of 0.91. These results indicate that the models are particularly effective for complex anatomical structures and enable significant time savings compared to manual segmentation. The three test bones have a neck Ct.Th between 2.21 and 2.55 mm, which is consistent with literature data (Carpenter et al., 2011; Treece et al., 2010). The volume of the femoral head is between 40.11 and 62.3 cm$^3$ and also corresponds to published values (Bugeja et al., 2022). No comparable literature data could be identified for the neck (27.56 - 32.85 cm$^3$) and trochanter (81.15 - 100.64 cm$^3$). The NSA is 119.94° to 121.94°, the FO is 51.33 to 56.09 mm. Due to differing definitions of the shaft axis, the NSA deviates slightly downwards and the FO upwards from literature values (Boese et al., 2016; Han et al., 2014; Yang et al., 2023; Li et al., 2024). The bone metrics were averaged over the ROIs of the head, neck and trochanter. BV/TV (0.23 - 0.34) and Tb.Th (0.30 - 0.37 mm) both tend to be above the literature values. Tb.Sp (0.53 - 0.85 mm) is slightly reduced, while Tb.N (0.90 - 1.16 mm$^{-1}$) is within the expected range (Fazzalari and Parkinson, 1996).

## 3. Conclusion, Limitations and Future Work

FemoraLyze is a modular framework for the automatic extraction of key bone metrics from the proximal femur, combining DL models with standardized analysis of multiple parameters. Its flexible architecture allows customizable processing units and selective module use, enabling adaptation to various clinical contexts such as prognosis, diagnosis, and therapy. The framework ensures time-efficient, reproducible evaluations and supports differentiated analyses through various segmentation masks. Despite the promising results, the present study is subject to some limitations. Comprehensive validation of the individual modules from both a technical and medical perspective is required for clinical application. In this context, the masks and characteristic reference points/regions are of particular importance, as all results depend significantly on them. In addition, the framework has so far only been tested on the proximal femur. This results in the following perspectives for future research: The results should be validated quantitatively and qualitatively by medical and technical experts. In addition, alternative methods could be implemented for individual modules and systematically compared in terms of effectiveness and efficiency. Finally, the system could be expanded to include additional modules and adapted to other bone structures.

**Acknowledgments.** Data was provided by Dr. med. Dr. rer. nat. F. von Brackel and Prof. Dr. med. B. Ondruschka from the University Medical Center Hamburg-Eppendorf (UKE).

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
