# OpenReview forum: "FemoraLyze: A Modular Framework for Proximal Femur Analysis"
_MIDL.io/2025/Short_Papers — MIDL 2025 - Short Papers_

### Official Review · Reviewer_BWHz · 2025-04-23

**Rating:** 3
**Confidence:** 4

**Summary:**

The authors propose a modular framework for the analysis of proximal femur. Their proposal is a modular deep-learning-based Python framework that combines the automation and calculation of segmentation masks, bone structure and geometry parameters from a computed tomography image.

**Strengths:**

- The manuscript is well written and structured.
- The work is clinically motivated and the results are compared with clinical literature.
- There is a discussion of limitations of their proposal. For example saying that the results should be validated quantitatively and qualitatively by medical and technical experts.

**Weaknesses:**

- The proposed work is not compared to other methods or frameworks working with femur CT data, lacking contextualization of the technical difficulties.
- The demographics in the dataset description are to vague.
- I suggest that the authors clarify in Figure 1 which steps are automated, which require human interaction, and where the human involvement occurs in the process.
- Code will only be available upon acceptance.

---

### Decision · Program_Chairs · 2025-05-01

Accept